# SARS-CoV-2 Serology: Utility and Limits of Different Antigen-Based Tests through the Evaluation and the Comparison of Four Commercial Tests

**DOI:** 10.3390/biomedicines10123106

**Published:** 2022-12-01

**Authors:** Mariem Gdoura, Habib Halouani, Donia Sahli, Mehdi Mrad, Wafa Chamsa, Manel Mabrouk, Nahed Hogga, Kamel Ben-Salem, Henda Triki

**Affiliations:** 1Faculty of Pharmacy of Monastir, University of Monastir, Monastir 5000, Tunisia; 2Laboratory of Clinical Virology, WHO Reference Laboratory for Poliomyelitis and Measles in the Eastern Mediterranean Region, Institut Pasteur de Tunis, University Tunis El Manar, Tunis 1002, Tunisia; 3Research Laboratory «Virus, Vectors and Hosts: One Health Approach and Technological Innovation for a Better Health» LR20IPT02, Institut Pasteur de Tunis, University Tunis El Manar, Tunis 1002, Tunisia; 4Clinical Investigation Center (CIC), Institut Pasteur de Tunis, University Tunis El Manar, Tunis 1002, Tunisia; 5Laboratory of Biochemistry and Hormonology, Institut Pasteur de Tunis, University Tunis El Manar, Tunis 1002, Tunisia; 6Department of Community Medicine, Faculty of Medicine of Monastir, University of Monastir, Monastir 5000, Tunisia; 7Réseau Maghrébin Pédagogie-Recherche-Publication (RPP2S), Monastir 5000, Tunisia

**Keywords:** SARS-CoV-2, serology, commercial tests, false positive, false negative

## Abstract

Introduction: SARS-CoV-2 serology have several indications. Currently, as there are various types available, it is important to master their performance in order to choose the best test for the indication. We evaluated and compared four different commercial serology tests, three of them had the Food and Drug Administration Emergency Use Authorization (FDA-EUA). Our goal was to provide new data to help guide the interpretation and the choice of the serological tests. Methods: Four commercial tests were studied: Elecsys^®^ Roche^®^ on Cobas^®^ (total anti-nucleocapsid (N) antibodies), VIDAS^®^ Biomerieux^®^ (IgM and IgG anti- receptor binding domain (RBD) antibodies), Mindray^®^ (IgM and IgG anti-N and anti-RBD antibodies) and Access^®^ Beckman Coulter^®^ (IgG anti-RBD antibodies). Two panels were tested: a positive panel (*n* = 72 sera) obtained from COVID-19-confirmed patients with no vaccination history and a negative panel (*n* = 119) of pre-pandemic sera. The analytical performances were evaluated and the ROC curve was drawn to assess the manufacturer’s cut-off for each test. Results: A large range of variability between the tests was found. The Mindray^®^IgG and Cobas^®^ tests showed the best overall sensitivity, which was equal to 79.2% CI 95% (67.9–87.8). The Cobas^®^ test showed the best sensitivity after 14 days of COVID-19 molecular confirmation; which was equal to 85.4% CI 95% (72.2–93.9). The Access^®^ test had a lower sensitivity, even after day 14 (55.5% CI 95% (43.4–67.3)). The best specificity was noted for the Cobas^®^, VIDAS^®^IgG and Access^®^ IgG tests (100% CI 95% (96.9–100)). The IgM tests, VIDAS^®^IgM and Mindray^®^IgM, showed the lowest specificity and sensitivity rates. Overall, only 43 out of 72 sera (59.7%) showed concordant results by all tests. Retained cut-offs for a significantly better sensitivity and accuracy, without significant change in the specificity, were: 0.87 for Vidas^®^IgM (*p* = 0.01) and 0.14 for Access^®^ (*p* < 10^−4^). The combination of Cobas^®^ with Vidas^®^ IgM and IgG offered the best accuracy in comparison with all other tests combinations. Conclusion: Although using an FDA-EUA approved serology test, each laboratory should carry out its own evaluation. Tests variability may raise some concerns that seroprevalence studies may vary significantly based on the used serology test.

## 1. Introduction

The severe acute respiratory syndrome coronavirus 2 (SARS-CoV-2) is an emerging virus that was first reported in December 2019 in Wuhan, China [1,2]. Rapidly, the virus spread across the globe and has become a major public health concern. On 11 March 2020, the World Health Organization (WHO) announced the COVID-19 disease as a pandemic [3]. To date, millions of infections by SARS-CoV-2 and hundreds of thousands of deaths have been attributed to COVID-19. Regular updates of new infections and deaths are available on the WHO COVID-19 dashboard. SARS-CoV-2 is a 80–120 nm spherical or pleomorphic enveloped particle. It is characterized by outer surface projected spike (S) proteins, along with membrane (M), envelope (E), nucleocapsid (N) and hemagglutinin (HA) proteins. S protein is characterized as a heavily glycosylated protein and contains the receptor binding domain (RBD), which is the most variable structure in coronaviruses that mediates viral entry into host cells [4]. Molecular testing by real-time PCR (RT-PCR) is the angular stone in the diagnosis of COVID-19. Since the beginning of the pandemic, it was playing a crucial role in testing, monitoring and contact tracing [5]. However, as screening indications are mainly limited to symptomatic patients, and given the result discrepancies between the RT-PCR tests, documented cases represent probably only the visible part of the iceberg [6]. For these reasons, other diagnostic methods were rapidly needed to better estimate SARS-CoV-2 spread [7,8]. Serology is ideally suited for this purpose as detecting specific anti-SARS-CoV-2 antibodies offers valuable information about previous contact with the virus, helps to assess the herd immunity at a large or specific population, and, recently, have had a decisive role to monitor vaccinated patients [9,10,11]. A worldwide laboratories and companies competition was launched soon after the virus emerged with the aim of developing efficient serology tests: they should have good sensitivity and specificity, be easy to use, give rapid results and have reasonable cost-effectiveness balance [12]. Today, many different tests are commercially available following different principles: enzyme-linked immune-sorbent assay (ELISA), enzyme-linked fluorescent assay (ELFA), eletrochimiluminescent assay (ECLIA), etc. These tests detect different isotypes: IgM, IgG, IgA or total antibodies and use different antigens: the full spike glycoprotein or sub-units S1 and S2, the receptor-binding protein (RBD) or nucleocapsid (N) [12,13,14,15,16,17]. In addition, many laboratories have developed and optimized their own in-house ELISA [18]. The choice of these target antigens was guided by a deep comprehension of the virus structure on the basis of in silico studies [19]. Fortunately, even after the emergence of the different variants of concern, these tests have preserved their capacity to detect antibodies, but for the interpretation of the results, it is necessary to know that very high titers of anti-RBD are necessary to have an optimal protection [20].

To facilitate their use on a global scale, international health authorities such as the Food and Drug Administration and the World Health Organization agreed to grant what they called an emergency use authorization (FDA-EUA) and an emergency use list (WHO-EUL), respectively [21,22,23]. That is why nowadays, serology has become widely used and can have many indications [24,25]. However, some tests may lack sufficient clinical evaluation, which made specialists establish their own evaluation and share concerns toward their performances [13,15,16,17].

In the present study, we aimed to evaluate the analytical performances of four different commercial serology tests and to compare them: Elecsys^®^ Roche^®^, VIDAS^®^ Biomerieux^®^, Mindray^®^ and Access^®^ Beckman Coulter^®^. The selected tests use different antigens (targeting either N, RBD or both of them), detect different isotypes (IgG, IgM or total antibodies), are based on different principles and three of them were authorized for an emergency use by the FDA.

Our goal was to provide experimental data which help to guide the choice of the serological tests according to the intended indication. Additionally, these data distinguish the utility of the different antigens targeted by each test (N and/or RBD).

## 2. Methods

### 2.1. Patients and Commercial Serology Tests

The study has been carried out in the Laboratory of Clinical Virology of Institut Pasteur de Tunis, Tunisia, during February and March 2021, and was approved by the institutional review board of the Institut Pasteur de Tunis (Approval number: 2020/27/I/LR16IPT). The selection of sera followed the guidelines of the French “*Centre National de Reference des Virus des Infections Respiratoires*” published on 4 December 2020 [26], i.e., the evaluation needs at least 50 true positive sera (a positive panel) and at least 50 true negative sera (a negative panel). The positive panel was composed of a total of 72 different unique, non-duplicated sera samples obtained from 72 COVID-19-confirmed patients on the basis of a positive RT-PCR on nasopharyngeal swab; all of them had no COVID-19 vaccination history and were obtained from May to December 2020. The studied patients were made of 44 female and 28 male, the sex ratio was equal to 1.6, the ages ranged from 11 to 76 with a median equal to 44 and a mean equal to 44 ė 14.2. Sera were collected from the first day (D0) until day 162 (D162) after molecular confirmation. They included 29 sera collected from D1 to D14, 16 sera collected from D16 to D30, 14 sera collected from D31 to D60 and 13 sera collected after D60 (until D162). The negative panel was composed of 119 pre-pandemic sera collected before 2019 and well-conserved at −20 °C. Four in vitro diagnosis (CE-IVD) commercial serology tests were evaluated: Elecsys^®^ Roche^®^ on Cobas^®^, Bale, Switzerland, FDA-EUA and WHO-EUL, based on ECLIA principle and detecting total anti-N antibodies (REF: 09203095119), VIDAS^®^ Biomerieux^®^, Marcy-l’Étoile, France, FDA-EUA, based on ELFA principle and detecting specific IgM and IgG anti-RBD antibodies (REF: 423833 and 423834, respectively), Mindray^®^, Shenzhen, China, based on CLIA principle and detecting specific IgM and IgG anti-N and anti-RBD antibodies (REF 105-019873-00-5 and 105-019883-00-5) and Access^®^ Beckman Coulter^®^, California, USA, FDA-EUA, based on CLIA principle and detecting specific IgG anti-RBD antibodies (REF C58961). All these tests are qualitative expressing results by ratios (index sera/cut-off index) and working on high throughput automated analyzers. The intrinsic characteristics and the manufacturer’s announced analytical performances of each test were summarized in Table 1.

### 2.2. Evaluation and Comparison

All sera from both panels (*n* = 191) were tested by each of the four tests, manipulations were carried out with strict respect to the manufacturers’ instructions. Each test was performed on its corresponding high throughput automated analyzer, provided by the local representative of each firm. All results were entered on a database sheet and were used to determine the overall sensitivity, the sensitivity after D14, the specificity, the positive predictive value (PPV), the negative predictive value (NPV) and the accuracy represented by the area under the curve (AUC). Based on the obtained ratios, a receiving operating characteristic (ROC) curve for each test has been drawn in order to assess the manufacturers’ cut-off, then to discuss better cut-offs. Finally, all tests’ results were compared and a combination of pooled results of different tests was statistically studied.

### 2.3. Statistical Analysis

All calculations and figures were performed using MEDCALC^®^V18.2.1 Ostend, Belgium. Using RT-PCR positive results as the reference test, sensitivity, specificity and AUC were calculated to assess the performance of each test. The T-test was used to compare the different obtained AUCs. The experimental results were used to draw the ROC curves for each test. The proposed Youden index J, which may help to display the discriminatory ability of a serological test to distinguish between true COVID-19 patients and non-previously infected patients, were discussed for each test. The significance level was set at 5% and a 95% confidence interval (CI 95%) was reported for each measure. As PPV and NPV are two prevalence-dependent analytical performances, their calculation was performed using the FDA calculator that proposed an arbitrary prevalence of 5%. This prevalence value was used by the FDA for the independent evaluations of COVID-19 serological tests to obtain the FDA-EUA [27].

## 3. Results

### 3.1. Evaluation of the Analytical Performances

The performances of each test were evaluated by calculating the sensitivity, specificity, PPV, NPV and AUC. All experimental results were shown in Table 2. The Mindray^®^IgG and Cobas^®^ tests showed the best overall sensitivity 79.2% CI 95% (67.9–87.8) (57 positive out of 72 true positive), but Cobas^®^ showed the best sensitivity after D14 85.4% CI 95% (72.2–93.9). For all IgG and total antibody tests, the sensitivity increased considerably after D14 except for Access^®^ (Table 2). Regarding the IgM tests, the VIDAS^®^IgM and Mindray^®^IgM tests showed the lowest sensitivity rates for all the sera including those collected after D14 (Table 2). The most accurate tests were Vidas^®^ IgG, Cobas^®^ and Mindray^®^ IgG (Table 2). For all tests, the accuracies followed the same improvements of the sensitivities over time.

Regarding the specificity, the best ones were noted for Cobas^®^, VIDAS^®^IgG and Access^®^ IgG: 100% CI 95% (96.9–100) (119 negative out of 119 true negative). Mindray^®^IgG was slightly less specific (95.8% CI 95% (90.5–98.6)). Indeed, positive results were obtained for 5 pre-pandemic patients: 1 having rheumatoid factors, 2 patients positive for *Herpes simplex virus* and 2 pregnant women; here, ratios ranged from 1.3 to 3.8. For IgM tests, VIDAS^®^IgM and Mindray^®^IgM tests showed the lowest specificity (Table 2). False positive tests were obtained for 7 pre-pandemic patients for Vidas^®^IgM: 3 patients having rheumatoid factors and 4 patients positive for *Herpes simplex virus*; the ratios ranged from 1.08 to 13.94. False positive tests were obtained for 3 pre-pandemic patients for Mindray^®^IgM: 2 patients having auto-immune disease and 1 patient positive for *Herpes simplex virus*; the ratios ranged from 1.94 to 4.97. The NPV was excellent for all tests; however, PPV lower than 50% were noted for Vidas^®^ IgM and Mindray^®^ IgM and IgG. PPV was excellent, however, for Vidas^®^ IgG, Cobas^®^ and Access^®^.

### 3.2. Improvement of the Analytical Performances

Using the ROC curve tool, we have attempted to adjust the cut-offs of the evaluated tests. All results were grouped in Table 2. Retained cut-offs for a significantly better sensitivity and accuracy while maintaining the same specificity rates were as follow: 0.87 instead of 1 for Vidas^®^IgM (*p* = 0.01) and 0.14 instead of 1 for Access^®^ (*p* < 10^−4^). The ROC curves for Vidas^®^IgM and Access^®^ using both cut-offs, the original and the proposed ones, are represented in Figure 1; they show the clear improvement of the tests’ accuracy. For Cobas^®^ Vidas^®^IgG and Mindray^®^IgM and IgG, the new proposed cut offs did not give better analytical performances than the original cut-offs (*p* > 0.05, Table 2), so they were rejected.

### 3.3. Antibody Ratio Distribution over Time

Figure 2 shows the scatter plots of the 4 tests’ ratios against days after COVID-19 confirmation. Figure 2A shows the distribution of the IgM tests ratios (Vidas^®^ and Mindray^®^), which was heterogeneous and does not fit a specific pattern; however, it was obvious that high indexes were obtained during the first 21 days after infection. Figure 2B illustrates the distribution of total antibodies and IgG antibodies tests indexes over time. It shows that antibodies are decreasing over time and no correlation with days after COVID-19 confirmation was found for Vidas^®^IgG, Mindray^®^IgG and Access^®^ (correlation coefficients were 0.001, 0.028 and 0.001, respectively). However, for the Cobas^®^, ratios are increasing over days with a correlation coefficient of 0.289.

### 3.4. Comparison of the Tests

The test accuracies were calculated and compared: Cobas^®^, VIDAS^®^IgG and Mindray^®^IgG had very good and similar accuracies (pairwise comparison of their respective ROC curves *p* > 0.05). However, Access^®^ had an accuracy of 0.778 CI 95% (0.712–0.835), which is good but, statistically, it was significantly lower than the other tests (*p* = 0.587). For the IgM tests, VIDAS^®^IgM and Mindray^®^IgM tests had overall good and similar accuracies (*p* = 0.587).

In order to dig into the origins behind the tests discrepancies, a detailed comparison per sera of the positive panel (*n* = 72) was performed as shown in Table 3. Overall, 43 out of 72 sera (59.7%) gave concordant results. Of them, 35 were positive, sampled between D4 and D140 and 8 were negative, sampled between D0 and D60. Discordant results represented 40.3% of the panel (29 out of 72). They were divided into 3 different patterns: the first pattern contained sera that were detected positive by 3 tests over 4 (*n* = 17); here, Access^®^ failed to detect 13 sera collected between D6 and D90 and Cobas^®^ failed to detect 4 sera collected between D8 and D39. The second pattern included sera that were detected positive by only 2 tests over 4 (*n* = 11) and the third pattern contained one sera that was detected by only one test over 4 which was Vidas^®^IgM and/or IgG for a sera collected at D7. It is worth noting that Mindray^®^ IgM and/or IgG test did not fail to detect any positive sera when the other tests did and was often concordant positive with at least one another test, the Vidas^®^ and/or Cobas^®^ in 17 cases out of 17 for the first discordant pattern and in 7 cases out of 11 for the second discordant pattern, with a total of 24 detected positive sera out of 29 total discordant sera.

### 3.5. Combinations between Tests

In order to improve diagnostic accuracy, different tests’ combinations were tried and the obtained accuracies (AUC of the ROC curves) were evaluated. For this purpose, the accuracy of Cobas^®^ (total anti-N antibodies) was compared with the accuracy of its combination with first, Vidas^®^ (anti-RBD, both isotypes IgM/IgG), and second, with Mindray^®^ (anti-N and anti-RBD, both isotypes IgM/IgG). No improvement of accuracies was found for these two combinations when compared to using only Cobas^®^ (pairwise comparison of ROC curves *p* > 0.05). These tests combinations were, therefore, unnecessary given the excellent relevance of using only Cobas^®^. We went further by evaluating the original antigen RBD and N combination, the Mindray^®^IgM/IgG with an equivalent combination made of Vidas^®^IgG/IgM with Cobas^®^, and the latter proved to be consistently more accurate than Mindray^®^IgM and IgG (pairwise comparison of ROC curves *p* = 0.0399).

## 4. Discussion

SARS-CoV-2 serology tests were developed and optimized in a record time after the virus emergence. Thanks to the softened authorization procedure, they were rapidly commercialized and used worldwide [21,22,23]. In this study, four serology commercial automated tests were evaluated and compared: Elecsys^®^Roche^®^ on Cobas^®^ (ECLIA) detecting total anti-N antibodies, mainly IgG, VIDAS^®^ Biomerieux^®^ (ELFA) detecting specific IgM and IgG anti-RBD antibodies, Mindray^®^ (CLIA) detecting specific IgM and IgG anti-N and anti-RBD antibodies and Access^®^ Beckman Coulter^®^ (CLIA) detecting specific IgG anti-RBD antibodies. Our evaluation revealed a gap between claimed and experimental analytical performances in terms of sensitivity and specificity and, accordingly, new analytical criteria were proposed. In addition, the comparison between the evaluated tests showed a significant divergence between the obtained qualitative results in 40.3% of the true positive tested sera (29 out of 72). Our findings suggest that the most sensitive test, after D14, was Cobas^®^ (85.4% IC 95%(72.2–93.9)), which detected high antibody ratios until 4 months after primo-infection. Besides, our study revealed that combining tests detecting anti-RBD and anti-N antibodies ensured the best diagnostic accuracy.

Overall, 72 RT-PCR-confirmed patients and 119 pre-pandemic sera were tested. Our work stands out from the rest of the literature by studying high throughput IVD commercial tests using different viral antigens and having internationally approved certificates, by proposing new significant cut-offs to improve the analytical performances and by a deep assessment of the origins behind discordances of the obtained results as well as the discussion of the utility and the limits of each antigen-based test. Thus, our work provided original and helpful data serving the health care professionals in their routine practice. Even though our panel was not too large, the number of tests was not big and the impact of the disease severity on the course of anti-SARS-CoV-2 antibodies kinetic was not assessed, our results are extendable given the representativeness of the panel (from D0 until D162), the diversity of the used tests (different antigens and isotypes) and the applicability of the conclusions for a routine practice in a clinical laboratory receiving all indications’ types.

The evaluation of commercial tests was widely reported for the SARS-CoV-2 virus as well as for other pathogens. International recommendations were published by several scientific societies and instances such as *Haute Autorité Sanitaire, France* HAS, FINDX *Foundation* for Innovative New Diagnostics, Public Health of England, UK and Health Canada in order to harmonize the criteria of validation of the tests [26,27,28,29,30]. In our study, the evaluation of all tests gave lower performances than the claimed ones and did not respond to the HAS validation criteria, the most flexible one, in terms of sensitivity (Table 1 and Table 2). Indeed, according to the HAS, the sensitivity of detecting IgG and total antibodies must exceed 90% after D14 from disease onset while for IgM antibodies, the sensitivity must exceed 90% after D7. It was reported that on the basis of a large review published by Cochrane on 15,976 sera, all tests showed low sensitivity, not exceeding 30.1% during the first week after the symptoms onset, then it rose in the second week to reach its highest values in the third week [31]. In our series, the best sensitivity after D14 was the one of Cobas^®^ (85.4% CI 95% (72.2–93.9)) followed by Vidas^®^IgG and Mindray^®^IgG (83.3% CI 95% (69.8–92.5)). Access^®^ came in the last position by a sensitivity of 55.5% CI 95% (43.4–67.3) (Table 2). For IgM detection, Mindray^®^ and Vidas^®^ had very low sensitivities even after D14. High sensitivity for Cobas^®^ found in this work corroborates the findings of other authors [13,16,17,32]. This could be explained by, first, the used antigen, which was exclusively the N protein, known to be the most abundantly expressed immune-dominant protein [24]; second, the ability of Cobas^®^ to detect all immunoglobulin isotypes (IgG, IgM and IgA), which was also reported for Siemens Atellica^®^ [33]; and third, the ECLIA Elecsys^®^ technology developed by Roche^®^, which was highly efficient regardless of the measured analyte [34,35]. For the other tests, such unsatisfactory sensitivities were reported by other studies for the same tests. Similar low sensitivities for the Vidas^®^ test were reported by Younes et al. (88.3% for Vidas^®^IgG after D21) and by Wolf et al. (over all sensitivity of 64.9% CI 95% (55.2–73.7) for Vidas^®^IgM and 73% CI 95% (63.7–81) for Vidas^®^IgG [36,37]. Padoan et al. also reported a sensitivity of 86.4 (77.0–93.0) for both Mindray^®^IgM and IgG tests but a new Mindray^®^ generation would give much better performances according to recent literature (99% and 96% from D1 to D41 for IgG and IgM, respectively) [38,39]. This new version of Mindray^®^ was not available at the study writing time and merits to be evaluated. Regarding the Access^®^ test, it showed very low sensitivity for IgG detection (55.5% CI 95% (43.4–67.3)) with no increasing trend after D14. Other authors reported similar poor sensitivities for Access^®^ such as Tan et al. by a value of 39.6% CI 95% (32.5–47.3%) [16,40]. Beckman^®^ has developed a new Access^®^ test version allowing anti-RBD semi quantification; it has issued FDA-EUA and, thus, merits to be evaluated in further studies.

Regarding the specificity, Vidas^®^IgM and Mindray^®^IgM and IgG tests gave positive signals for few pre-pandemic sera (n = 7); such data were also reported by other authors [36,38,39]. Cross reactivity with pre-pandemic auto-immune disease patients sera was previously reported [41]. In contrast, cross reactivity with pre-pandemic pregnant women sera, and patients positive for the *Herpessimplex virus* was reported for the first time in the present paper. Further characterization of these sera should be performed in order to depict the responsible epitope of the cross reactivity. Meanwhile, results of Vidas^®^IgM and Mindray^®^IgM and IgG should be interpreted with caution. Indeed, PPV of these tests were lower than 50%, which means that half of the tested patients are susceptible to be a false positive. Here, PPV and NPV were calculated by the FDA calculator fixing the prevalence at 5%. However, each country is invited to evaluate regularly the PPV according to the real-time prevalence evolution. Cobas^®^, Access^®^ and Vidas^®^ IgG were extremely specific tests (100% IC 95% (96.9–100)); their PPV was 100%, which corroborates data reported by previous studies [17,32,36].

The analysis of sensitivity, specificity, PPV and NPV was based on the interpretation criteria of the different manufacturers. However, for the purposes of this study, the pre-defined tests’ cut-offs were experimentally optimized and adjusted for an improved sensitivity with very little loss in specificity. This approach is being widely used and reported by many authors for better interpretation of commercial tests, for COVID-19 tests as well as other pathogens [36,42,43]. Our findings suggest that decreasing the cut-off signals for Vidas^®^IgM and Access^®^ improved significantly the sensitivity as well as the accuracy (Table 3). As none of the tests proposed a grey zone for borderline results, which is unusual for the interpretation of low signals in routine practice, we recommend that all results belonging to the range (proposed cut-off—original cut-off) (i.e., (0.87 to 1) for Vidas^®^IgM and (0.14–1) for Access^®^) should be retested and if possible, the patient should be re-sampled after 10 to 15 days to follow the antibody kinetic. More generally, any weak signal, i.e., lower than two times the manufacturer’s cut-off, should be interpreted with caution.

Comparison between the four tests showed concordant results in 59.7% of the sera collected in confirmed cases (43 out of 72) among which, 35 were positive by all tests and 8 were negative by all tests. These 8 sera were collected between D0 and D60, and the median was equal to 14. Here, as none of the four different tests could detect antibodies, though using different antigens, this may be inherent to the individuals’ immune system. In fact this may be explained by either a late sero-conversion or a rapid sero-reversion [44]. Moreover, some authors had suggested that 5 to 10% of the SARS-CoV-2-infected patients may not develop antibodies at all [45]. Regarding the discordant group, a non-negligible proportion of sera was noted (40.3%, *n* = 29 out of 72). Access^®^ was the test that failed the most to detect positive results (13 cases out of 29 discordant results). For the three other tests, various discordant patterns were regrouped in Table 3. Among the discordant patterns, 24 out of 29 sera were positive by Mindray^®^ which underline that a combination of N and RBD antigens would increase the number of true positive sera. Indeed, although Mindray^®^ is not an FDA-EUA test, it was introduced in this study for its originality, as it is multiplex (N and RBD). Our data demonstrated that Mindray^®^IgM and IgG offers a similar good sensitivity as Cobas^®^, but the combination made of Cobas^®^ with Vidas^®^IgM/IgG exceeds the combination made of Mindray^®^ IgM/IgG in accuracy. So, a two-step strategy starting by testing Cobas^®^ then Vidas^®^IgM/IgG could improve significantly the sensitivity, and offers separate comprehension of antibodies specificity.

Questions regarding the magnitude and the longevity of the antibody response remain unanswered. Many literature reviews tried to propose a general kinetic of antibodies and recognize a big variability between individuals and proportionality with COVID-19 severity [46,47]. In our study, Figure 2A showed that scatter plots of the two IgM tests were high and condensed among the first 3 weeks, suggesting that their detection was in line with an ongoing or acute infection. However, in this study, it was found that IgM may be still detectable even until D162. Regarding IgG, (Figure 2B) anti-RBD antibodies are decreasing over time, with no significant correlation with days after COVID-19 molecular confirmation and following almost the same decay for both Vidas^®^ and Acces^®^. This is in contrast with anti-N antibodies (Cobas^®^) that continue to be positive with high ratios for longer time. This may be explained by half-life time for IgG anti-RBD antibodies, which is 49 days versus 75 days for IgG anti-N antibodies [47].

Finally, our data may be helpful for clinicians, biologists and researchers. They are providing some keys for clinicians to adapt the serology test to the indication, especially when serology is used as a second line of diagnosis after molecular tests. In addition, our finding represent some helpful tools to the biologists to interpret the obtained results in a critical manner and guide the choice of new tests to implement and even, to regularly evaluate the already laboratory acquired tests. As well, the present paper may offer some recommendations for public health researchers and epidemiologists to select the best high throughput test that helps determining the seroprevalence of the SARS-CoV-2.

In conclusion, it is true that serological tests do not replace molecular tests in diagnosing active infection, but they are multipurpose and provide the choice for the most efficient test and to properly interpret the obtained results. The performance of four commercial serology platforms used worldwide was studies, and the variability between them was detected and explained. Although FDA approved, each laboratory should realize its own evaluation for commercial tests, using a larger sample for better results. Health professionals should be aware about the false negative rate before 14 to 21 days after primo-infection. Finally, this variability may raise some concerns that seroprevalence studies may vary significantly based on the used serology test.

## Figures and Tables

**Figure 1 biomedicines-10-03106-f001:**
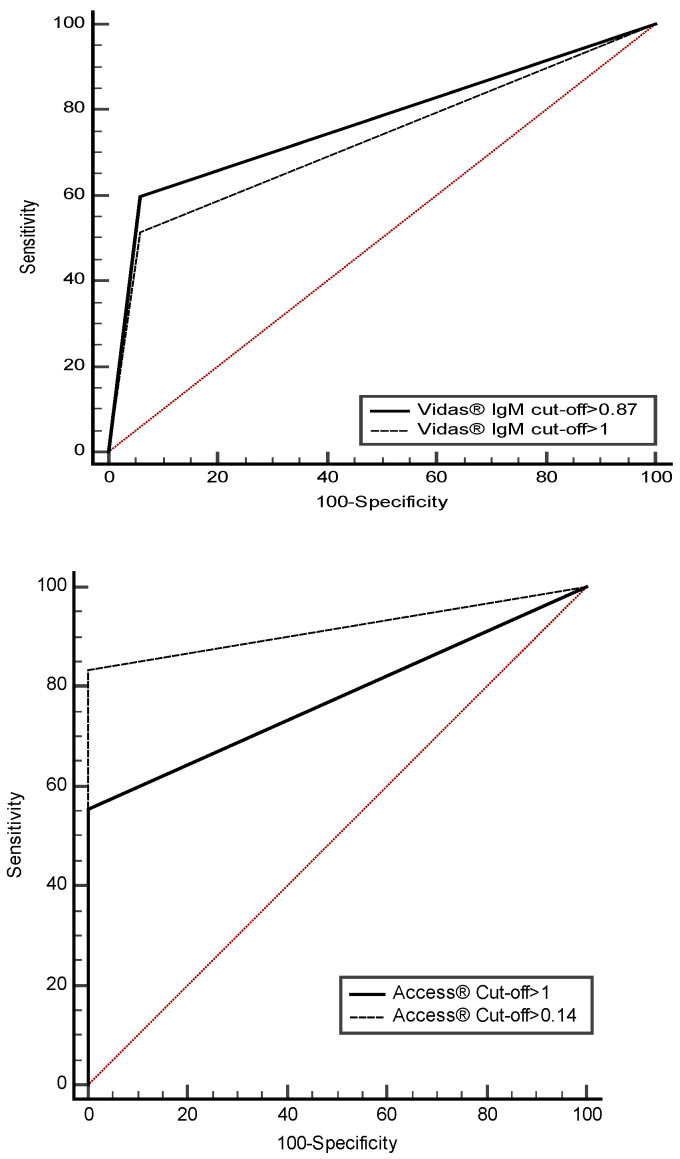
ROC curves of Vidas^®^ IgM test (at the top) and Access^®^ test (at the bottom) using the original cut-offs (continuous line) and the proposed cut-offs (discontinuous line), *p* < 0.05, diagonal line shows equality.

**Figure 2 biomedicines-10-03106-f002:**
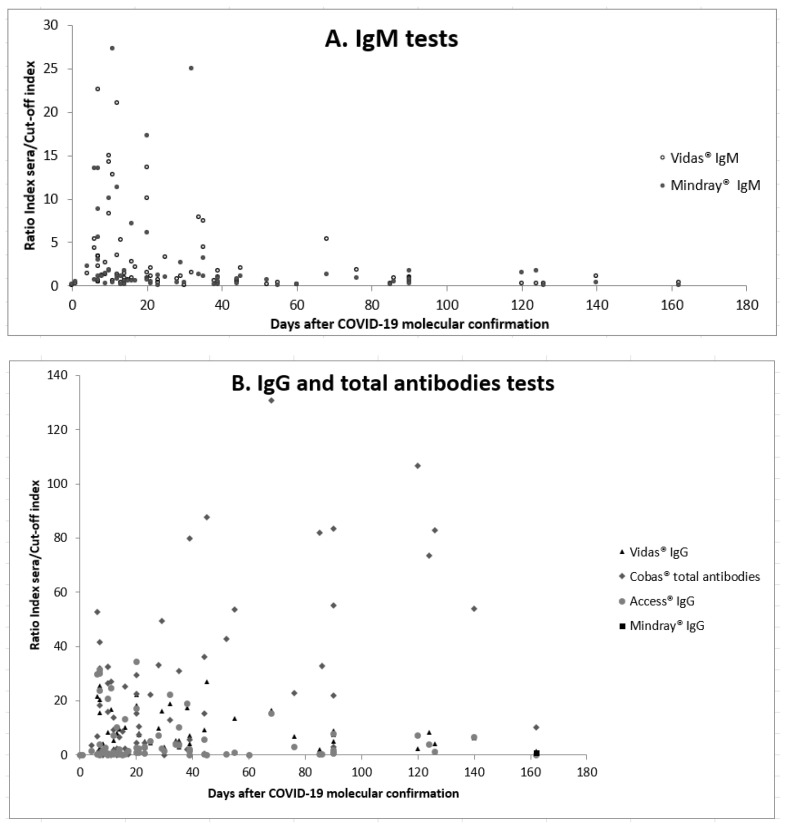
Scatter plots of the 4 tests’ ratios against days after COVID-19 confirmation (**A**), IgM tests, (**B**), IgG and total antibodies tests.

**Table 1 biomedicines-10-03106-t001:** Characteristics of the four evaluated commercial serology tests according to the manufacturer’s prospectus.

	Certifications	Detection Principle	Antibodies Isotypes	Targeted Antigen	Sample Volume	Cut-Off	Result Interpretation	Reported Sensitivity	Reported Specificity
Vidas^®^	CE-IVD, FDA-EUA	ELFA, qualitative	IgM IgG	S1 RBD	100 μL including the dead volume	1 for both	Index <cut-off; Negative Index > or = cut-off; Positive	IgM 100% CI 95% (63.1–100.0) after 8 days IgG 100% CI 95% (80.5–100.0) after 15 days	IgG 99.9% CI 95% (99.4–100.0) IgM 99.4% CI 95% (97.7–99.9)
Mindray^®^	CE-IVD	CLIA, qualitative	IgM IgG	S and N protein	10 μL with a minimum of 100 μL of dead volume	1 for IgM10 for IgG	Index < cut-off; Negative Index > or = cut-off; Positive	IgM 82.22% after 15 days IgG 100% after 15 days	87.60% 94.9%
Cobas^®^	CE-IVD, FDA-EUA, WHO-EUL	ECLIA, qualitative	Total antibodies: IgG+++, IgM and IgA	N protein	10 μL with a minimum of 100 μL of dead volume	1	Index <cut-off; Negative Index > or = cut-off; Positive	100% CI 95% (88.1–100) after 14 days	99.8% CI 95% (99.7–99.9)
Access^®^	CE-IVD, FDA-EUA	CLIA, qualitative	IgG	S1 RBD	10 μL with a minimum of 100 μL of dead volume	1	Index < cut-off; Negative Index > or = cut-off; Positive	100% CI 95% (93.8–100) after 18 days	99.8% CI 95% (99.4–99.9)

**Table 2 biomedicines-10-03106-t002:** Experimental analytical performances for the four used commercial tests for SARS-CoV-2 antibodies detection: according to the manufacture’s criteria and following the new proposed cut-offs.

	Experimental Data							Data Based on Proposed Cut-Offs			
	Sensitivity ≥ 14 Days % CI 95%	Sensitivity % CI 95%	Specificity % CI 95%	AUC CI 95%	PPV % CI 95%	NPV % CI 95%	Proposed Cut-Off	Sensitivity% CI 95%	Sensitivity ≥ 14 days% CI 95%	Specificity% CI 95%	AUC CI 95%
Vidas^®^ IgM	39.6 (25.7–54.7)	51.4 (39.3–63.3)	94.1 (88.2–97.6)	0.728 (0.659–0.789)	31.5 (17.8–49.4)	97.4 (96.6–97.9)	>0.87, *p* = 0.01	59.7 (47.5–71.1)	52 (37.2–66.7)	94.1 (88.3–97.6)	0.767 (0.703–0.827)
Vidas^®^ IgG	83.3 (69.8–92.5)	76.4 (64.9–85.6)	100 (96.9–100)	0.882 (0.828–0.924)	100	98.7 (98.1–99.2)	>0.55, *p* = 0.05	84.7 (74.3–92.1)	91.2 (80–97.7)	98.3 (94.1–99.8)	0.922 (0.875–0.956)
Cobas^®^	85.4 (72.2–93.9)	79.2 (68–87.8)	100 (96.9–100)	0.896 (0.844–0.935)	100	98.9 (98.3–99.3)	>0.725, *p* = 0.36	81.94 (71.1–90)	85.4 (72.2–93.9)	99.2 (95.4–100)	0.906 (0.855–0.943)
Access^®^	41.7 (27.6–56.8)	55.5 (43.4–67.3)	100 (96.9–100)	0.778 (0.712–0.835)	100	97.7 (97–98.2)	>0.14, *p* < 10^−4^	83.3 (72.7–91.1)	83.3 (69.8–92.5)	100 (96.9–100)	0.917 (0.868–0.952)
Mindray^®^ IgM	66.7 (51.6–79.6)	44.4 (32.7–56.6)	97.5 (92.8–99.5)	0.710 (0.640–0.773)	48.1 (22.7–74.5)	97 (96.4–97.6)	>0.83, *p* = 0.63	47.2 (35.3–59.3)	37.5 (23.9–52.6)	95.8 (90.5–98.6)	0.715 (0.645–0.778)
Mindray^®^ IgG	83.3 (69.7–92.5)	79.2 (68–87.8)	95.8 (90.5–98.6)	0.875 (0.819–0.918)	49.8 (29.4–70.2)	98.8 (98.2–99.3)	>7.94, *p* = 0.34	81.9 (72.7–91.9)	87.5 (74.7–95.2)	94.1 (88.3–97.6)	0.887 (0.834–0.928)

**Table 3 biomedicines-10-03106-t003:** Concordance and discordance results obtained by the four evaluated test results for the positive panel (*n* = 72).

	Group	VIDAS^®^ IgG and/or IgM	Cobas^®^	Access ^®^	Mindray ^®^ IgG and/or IgM	*n*	*n* days after COVID-19 Infection Med(Min-Max)
Concordant results between tests (43 out of 72, 59.7%)	All tests positive	+	+	+	+	35	25 (4–140)
	All tests negative	-	-	-	-	8	14 (0–60)
Discordance results between tests (29 out of 72, 40.3%)	Pattern 1: 3 positive tests over 4 *n* = 17	+	+	-	+	13	21 (6–90)
		+	-	+	+	4	20 (8–39)
		-	+	+	+	0	NA
	Pattern 2: 2 positive tests over 4 *n* = 11	+	+	-	-	3	39, 90, 162
		+	-	+	-	1	17
		+	-	-	+	1	9
		-	+	+	-	0	NA
		-	+	-	+	6	23 (15–86)
		-	-	+	+	0	NA
	Pattern 3: 1 positive test over 4 *n* = 01	+	-	-	-	1	7
		-	+	-	-	0	NA
		-	-	+	-	0	NA
		-	-	-	+	0	NA

Med: median, Min: minimum, Max: maximum, NA: not applicable.

## Data Availability

The data that support the findings of this study are available from the corresponding author upon reasonable request.

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
