# Peer review of "SARS-CoV-2 Serology: Utility and Limits of Different Antigen-Based Tests through the Evaluation and the Comparison of Four Commercial Tests"

_biomedicines, 2022, doi:10.3390/biomedicines10123106_

Round 1

Reviewer 1 Report

The manuscript is well written and the results supporting the conclusions.

I think the manuscript is providing enough data to support their conclusion. There are many aspects of the serology assay which can be evaluated for more accurate unbiased test results and choice of the assay. 

For example:

1. age group and gender differentiation influence the test results.

2.  How comparable are the results of the individual serology assay with RT-PCR of the samples?

3. In this study, the authors used only 72 samples as a positive panel. Increase in number may provide more accuracy of the assay results. 

4. More controls can be evaluated to assess the accuracy of the assay. But  the limitation is already determined by the assay developing company. The authors can not change that. Authors already mentioned in the discussion that variability is an issue among the used serology tests. 

But those questions may be out of the scope of the study. The manuscript is only providing the data to evaluate the efficiency of the serology test available in the market. Table 2 provides the cut-off value which is relatively new  data and important for the determination of the accuracy. Addition of more samples to the test evaluation may increase the accuracy of detection determinants.    The only thing may need to be revised but not essential. The discussion is too lengthy and missing the connection between paragraph 3 and 4 (page 8). 

Author Response

biomedicines-1979425

Dear Reviewer 1,

We are very grateful to you for agreeing to review our manuscript entitled « SARS-CoV2 serology assays: utility and limits of different antigen based tests through the evaluation and the comparison of four commercial tests ». We would like to thank you for careful and thorough reading of this manuscript and for your precious comments and constructive suggestions, which helped to improve the quality of our manuscript. Below are our responses given in a point-by-point manner; all your comments were considered and the manuscript was revised accordingly. The revised manuscript is submitted electronically and the corrections were highlighted with 'tracked changes'. We believe that things are clearer for the reader in the revised version of the manuscript. We hope that you will now find our manuscript acceptable for publication in your honorable Journal.

Yours faithfully,

The corresponding author

Prof. Mariem Gdoura

Comments and Suggestions for Authors

The manuscript is well written and the results supporting the conclusions.

I think the manuscript is providing enough data to support their conclusion. There are many aspects of the serology assay which can be evaluated for more accurate unbiased test results and choice of the assay. 

For example:

  1. age group and gender differentiation influence the test results.

=>We agree with you that age group and gender differentiation influence the test results. For the purpose of this study, we followed the guidelines of the French "Centre Nationalde Referencedes Virus des Infections Respiratoires "publishedonDecember 4th2020 which recommended as unique criterion to have a previous positive SARS-CoV2 RT-PCR test, and with this we selected patients belonging to both genders and different age groups. The age and sex were added in the revised manuscript (Lines 146-148, Methods Section).

  1. How comparable are the results of the individual serology assay with RT-PCR of the samples?

=>Indeed, the patients who were selected in the positive panel all had a positive RT-PCR test, at more or less different times, it is the reference test that allowed us to calculate the sensitivity of the evaluated tests. All values of overall sensitivity were represented in table 2.

  1. In this study, the authors used only 72 samples as a positive panel. Increase in number may provide more accuracy of the assay results. 

=>We agree that the higher the number of sera the better the comparison between tests, however, referring to recommendations quoted in the methodology, we increased by almost half of what was recommended. In addition, for an unbiased evaluation, we have selected patients who have been infected only once with the virus (from May to December 2020) and not vaccinated, something that is very difficult to find nowadays.

  1. More controls can be evaluated to assess the accuracy of the assay. But  the limitation is already determined by the assay developing company. The authors can not change that. Authors already mentioned in the discussion that variability is an issue among the used serology tests. But those questions may be out of the scope of the study. The manuscript is only providing the data to evaluate the efficiency of the serology test available in the market. Table 2 provides the cut-off value which is relatively new  data and important for the determination of the accuracy. Addition of more samples to the test evaluation may increase the accuracy of detection determinants.    The only thing may need to be revised but not essential.

The discussion is too lengthy and missing the connection between paragraph 3 and 4 (page 8). 
=> A linking sentence was added to the discussion before the paragraph 4 (Lines 399-400, Discussion Section)

Reviewer 2 Report

Title; SARS-CoV2 serology assays: utility and limits of different antigen based tests through the evaluation and the comparison of four commercial tests
Comments; In my view, the results obtained in this study are worthy for publication. The manuscript needs major essential revision before publication. I would like to overview the revised version of the manuscript. I have the following comments/suggestions for authors to address before final decision on the manuscript.
1. “SARS-CoV2” should be written as SARS-CoV-2.
2. “SARS-CoV2 serology testing is multipurpose provided to choose an efficient test.”: Re-frame the sentence.
3. What was the rationale/logic for the selection of four tests only? This should be highlighted in the Introduction.
4. “Our goal was to provide experimental data which help to guide the choice of the serological tests according to their indication, as well as the interpretation of the serology profiles on the basis of the used antigens.”: What do authors mean by “interpretation of the serology profiles on the basis of the used antigens”?
5. “COVID-19 dashboard as of 15 July 2022, the total of infections and deaths numbers were 557 917 904 and 6 358 899, respectively”: Avoid using the data being updated on a daily basis.
6. Clearly define the aim and objectives of the study in the last paragraph of the Introduction section.
7.
In the Introduction section the author should refer to the research paper and comment on recent in-silico techniques. It will be good information for the readers. I would like to recommend several papers, among many others, providing further explanation on this topic: PMID: 34099976 PMID: 34273770 PMID: 32397940 PMID: 34717229 PMID: 34136511 PMID: 34144270 PMID: 33217661 PMID: 33418408 PMID: 35551011 PMID: 35908093 PMID: 35896605 PMID: 36203381
8. The results are based on a relatively low number of test subjects. Authors should mention this as a limitation of the study in the Discussion as well as the Conclusions section.
9. Incorrect way of writing “Mindray®IgG and Cobas® tests showed the best overall sensitivity 79,2%CI95%[67,9-87,8]. Cobas® showed the best sensitivity after day 14; 85,4%CI95%[72,2 93,9]. Access® had the lower sensitivity even after day 14 (55,5% 28 CI95%[43,4-67,3]). VIDAS®IgM and Mindray®IgM tests showed the lowest specificity and sensitivity rates. The best specificity was noted for Cobas®, VIDAS®IgG and Access® 30 IgG(100%CI95%[96,9-100]).”
10. Authors have not provided structural data of SARS-CoV-2 in the introduction section.
11. Authors have written “In the present study we conducted an evaluation and a head-to-head comparison of 4 different commercial serology tests targeting either the N or the RBD protein or both of them; three of them had an FDA-EUA” Provide the serological test and clarify N and RBD protein.
12. Method section lacks proper referencing, the authors have to add the relevant references.
13. Representation of figure 2 should be changed and the resolution of the image needs to be increased.
14. Authors have advised to incorporate the limitations and future aspects of the analysis in the discussion section.
15. “That’s why, nowadays, serology has become widely used with many indications (22, 23).” The meaning is not clear.
16. “sensitivity and accuracy–with no decrease in specificity-were:” Punctuation error in the line.
17. “Over time Antibodies are decreasing over time and no correlation with D was found” Repetition of over time in the sentence. Authors also mention what this D signifies here.
18. Will these tests be affected by new variants of COVID. If not, a little highlights should be given for the same in the introduction section.
19. “Our findings suggest that the most sensitive test, after D14 was Cobas® (85,4%IC95%[72,2-93,9])” I thought it must be CI value instead of IC. Authors should correct it.

Author Response

biomedicines-1979425

Dear Reviewer 2,

We are very grateful to you for agreeing to review our manuscript entitled « SARS-CoV2 serology assays: utility and limits of different antigen based tests through the evaluation and the comparison of four commercial tests ». We would like to thank you for careful and thorough reading of this manuscript and for your precious comments and constructive suggestions, which helped to improve the quality of our manuscript. Below are our responses given in a point-by-point manner; all your comments were considered and the manuscript was revised accordingly. The revised manuscript is submitted electronically and the corrections were highlighted with 'tracked changes'. We believe that things are clearer for the reader in the revised version of the manuscript. We hope that you will now find our manuscript acceptable for publication in your honorable Journal.

Yours faithfully,

The corresponding author

Prof. Mariem Gdoura

Comments and Suggestions for Authors

Title; SARS-CoV2 serology assays: utility and limits of different antigen based tests through the evaluation and the comparison of four commercial tests
Comments; In my view, the results obtained in this study are worthy for publication. The manuscript needs major essential revision before publication. I would like to overview the revised version of the manuscript. I have the following comments/suggestions for authors to address before final decision on the manuscript.
1. “SARS-CoV2” should be written as SARS-CoV-2.

=>All SARS-CoV2 words were replaced by SARS-CoV-2
2. “SARS-CoV2 serology testing is multipurpose provided to choose an efficient test.”: Re-frame the sentence.

=>This sentence was reframed (Lines 16-18, Abstract Section)
3. What was the rationale/logic for the selection of four tests only? This should be highlighted in the Introduction.

=>We chose tests that are using different antigens, detecting different isotypes, based on different principle and 3 of them had international validations such as the FDA-EUA. This allowed us to perform a deep comparison. A sentence in the introduction was added to underline this idea (Lines 97-99, Introduction Section)
4. “Our goal was to provide experimental data which help to guide the choice of the serological tests according to their indication, as well as the interpretation of the serology profiles on the basis of the used antigens.”: What do authors mean by “interpretation of the serology profiles on the basis of the used antigens”?

=>The mean of this sentence is that a positive serology of anti-RBD test means the potential presence of an immunity against the virus, because anti RBD are correlated to a neutralizing potential, however, a positive serology of anti-N test indicates a previous contact with the virus, no conclusion about the immunity can be obtained. Few sentences were added to clarify this idea. (Lines 101-103, Introduction Section)
5. “COVID-19 dashboard as of 15 July 2022, the total of infections and deaths numbers were 557 917 904 and 6 358 899, respectively”: Avoid using the data being updated on a daily basis.

=>This sentence was deleted and replaced by a more general sentence. (Line 57, Introduction Section)
6. Clearly define the aim and objectives of the study in the last paragraph of the Introduction section.
=>The last paragraph of the introduction section was modified in order to make our objectives clearer for the reader. (Lines 95-103, Introduction Section)

7.In the Introduction section the author should refer to the research paper and comment on recent in-silico techniques. It will be good information for the readers. I would like to recommend several papers, among many others, providing further explanation on this topic: PMID: 34099976 PMID: 34273770 PMID: 32397940 PMID: 34717229 PMID: 34136511 PMID: 34144270 PMID: 33217661 PMID: 33418408 PMID: 35551011 PMID: 35908093 PMID: 35896605 PMID: 36203381

=> Few sentences were added to underline the utility of the in silico studies as well as a new reference PMID: 34273770 (Lines 82-83, Introduction Section)
8. The results are based on a relatively low number of test subjects. Authors should mention this as a limitation of the study in the Discussion as well as the Conclusions section.

=> A sentence was added in the Discussion section (Line 313) and conclusion section (Line 471).
9. Incorrect way of writing “Mindray®IgG and Cobas® tests showed the best overall sensitivity 79,2%CI95%[67,9-87,8]. Cobas® showed the best sensitivity after day 14; 85,4%CI95%[72,2 93,9]. Access® had the lower sensitivity even after day 14 (55,5% 28 CI95%[43,4-67,3]). VIDAS®IgM and Mindray®IgM tests showed the lowest specificity and sensitivity rates. The best specificity was noted for Cobas®, VIDAS®IgG and Access® 30 IgG(100%CI95%[96,9-100]).”

=> These sentences were modified in a clearer way (Lines 28-33, Abstract Section)
10. Authors have not provided structural data of SARS-CoV-2 in the introduction section.

=>Few sentences were added about the structure of the SARS-CoV2 (Lines 58-63, Introduction Section)
11. Authors have written “In the present study we conducted an evaluation and a head-to-head comparison of 4 different commercial serology tests targeting either the N or the RBD protein or both of them; three of them had an FDA-EUA” Provide the serological test and clarify N and RBD protein.

=>The serological tests were added and the mean of N and RBD, as well (Lines 96-100, Introduction Section)
12. Method section lacks proper referencing, the authors have to add the relevant references.

=>Appropriate reference for the followed guidelines is number 24 “Theselectionof sera followed the guidelines of the French “Centre Nationalde Referencedes Virus des Infections Respiratoires”publishedonDecember 4th2020(24)”. Regarding the used commercial tests, we mentioned the exact reference of the test that exist in the package (Lines 153-160).
13. Representation of figure 2 should be changed and the resolution of the image needs to be increased.

=>The Figure 2 was replaced by another one with better resolution
14. Authors have advised to incorporate the limitations and future aspects of the analysis in the discussion section.
15. “That’s why, nowadays, serology has become widely used with many indications (22, 23).” The meaning is not clear.

=> We mean that these tests are used with various indications: to make a late COVID-19 diagnosis, to reveal a previous asymptomatic indication, to evaluate the presence of a good immunity… The sentence was reframed to make it clearer (Line 92, Introduction Section)
16. “sensitivity and accuracy–with no decrease in specificity-were:” Punctuation error in the line.

=> The punctuation error was corrected and the sentence was slightly corrected in the revised manuscript (Line 229-230, Result Section)
17. “Over time Antibodies are decreasing over time and no correlation with D was found” Repetition of over time in the sentence. Authors also mention what this D signifies here.

=>The sentence was reframed and the meaning of D was clarified (Line 247-248, Result Section)
18. Will these tests be affected by new variants of COVID. If not, a little highlights should be given for the same in the introduction section

=>According to literature, the ability of these tests to detect antibodies seem to be not affected by variants, however, to be immunized against variants, higher titers are needed. A sentence with appropriate reference were added (lines 84-87, Introduction Section)
19. “Our findings suggest that the most sensitive test, after D14 was Cobas® (85,4%IC95%[72,2-93,9])” I thought it must be CI value instead of IC. Authors should correct it.

=>IC was replaced by CI (Line 309, Discussion Section)
